# SARS-CoV-2 Variants Identification; A Fast and Affordable Strategy Based on Partial S-Gene Targeted PCR Sequencing

**DOI:** 10.3390/v14112588

**Published:** 2022-11-21

**Authors:** Antonio Martínez-Murcia, Adrian Garcia-Sirera, Aaron Navarro, Laura Pérez

**Affiliations:** 1Department of Microbiology, University Miguel Hernández, 03312 Orihuela, Spain; 2Genetic PCR Solutions™, 03300 Orihuela, Spain; technic_ags@geneticpcr.com (A.G.-S.); research@geneticpcr.com (A.N.); gps@geneticpcr.com (L.P.)

**Keywords:** SARS-CoV-2, variants, lineages, S gene, PCR, sequencing

## Abstract

A considerable number of new SARS-CoV-2 lineages have emerged since the first COVID-19 cases were reported in Wuhan. As a few variants showed higher COVID-19 disease transmissibility and the ability to escape from immune responses, surveillance became relevant at that time. Single-nucleotide mutation PCR-based protocols were not always specific, and consequently, determination of a high number of informative sites was needed for accurate lineage identification. A detailed in silico analysis of SARS-CoV-2 sequences retrieved from GISAID database revealed the S gene 921 bp-fragment, positions 22784–23705 of SARS-CoV-2 reference genome, as the most informative fragment (30 variable sites) to determine relevant SARS-CoV-2 variants. Consequently, a method consisting of the PCR-amplification of this fragment, followed by Sanger’s sequencing and a “single-click” informatic program based on a reference database, was developed and validated. PCR-fragments obtained from clinical SARS-CoV-2 samples were compared with homologous variant-sequences and the resulting phylogenetic tree allowed the identification of Alpha, Delta, Omicron, Beta, Gamma, and other variants. The data analysis procedure was automatized and simplified to the point that it did not require specific technical skills. The method is faster and cheaper than current whole-genome sequencing methods; it is available worldwide, and it may help to enhance efficient surveillance in the fight against the COVID-19 pandemic.

## 1. Introduction

The Severe acute respiratory syndrome coronavirus-2 (SARS-CoV-2) is an enveloped, positive sense, single-stranded RNA virus. It belongs to the *Betacoronaviriae* genus, and it was first detected in China in December 2019. As it is generally known, RNA viruses show relatively high mutational rates [1,2,3,4], and despite the proof reading-repair 3′ to 5′exonuclease mechanism present in some coronaviruses [5,6,7,8], SARS-CoV-2 acquires mutations in each replication cycle, giving rise to new lineages. According to the definition given by Centers for Disease Control and Prevention (CDC) in Atlanta (USA), a new lineage is considered as a group of viruses closely related sharing a common ancestor (https://www.ecdc.europa.eu/en/COVID-19/variants-concern, accessed on 10 October 2022). All the lineages described to date (October 2022) may cause COVID-19, and many of them have been referred to as variants of concern (VOC), variant of interest (VOI), or variant being monitored (VBM) [9] due to the acquisition of mutations associated. For example, changes on the receptor binding domain are expected to affect functional properties, altering infectivity capacity, disease severity, and/or interactions with host immunity leading to immune escape [10,11,12,13]. Since the beginning of the pandemic, and with the first reports of the SARS-CoV-2 isolate Wuhan-Hu-1 (GenBank accession no: MN908947) [14], hundreds of new variants have been described in different geographic regions [10,11,15].

Omicron (lineages B.1.1.529 and BA.x) and Delta (lineages B.1.617.2 and AY.x) were the only two variants considered as VOC at the time of writing this article, while variants Alpha (lineages B.1.1.7 and Q.x), Beta (Lineages B.1.351.1, B.1.351.2 and B.1.351.3), Gamma (Lineage P.1 and P.1.x), Epsilon (lineages B.1.427 and B.1.429), Eta (lineage B.1.525), Iota (lineage B.1.526), Kappa (lineage B.1.617.1), Zeta (lineage P.2), Mu (lineages B.1.621 and B.1.621.1), Lambda (C.37 and C.37.1), and lineage B.1.617.3 are now considered as variants being monitored (VBM) because their expansion has been recently diminished and controlled to a great extent. SARS-CoV-2 genome evolution has been analysed since 2020, and several sets of mutations have emerged in the context of variants that impact virus characteristics, including transmissibility and antigenicity [10,13,16]. Most mutations determining VOC or VOI are affecting the spike protein (encoded by the S gene), a primary antigen of SARS-CoV-2 [17] related to infective capacity and immunological responses [13,18,19].

Lineage identification has been addressed in several methods, mainly strategies based on PCR targeting single or very few nucleotide mutations [20]. However, single-nucleotide mutations (SNPs) are not always SARS-CoV-2 variant-specific, and a large number of SNPs need to be typed for reliable identification of each lineage. Whole-genome sequencing (WGS) has been considered as the gold-standard method to determine SARS-CoV-2 variants; nevertheless, this technology is complex and expensive, not affordable for most laboratories. Since the beginning of 2021, an action protocol named Spike Gene Target Failure (SGTF) was extended to identify the Alpha variant (B.1.1.7). According to this protocol, a negative result obtained from the PCR targeting a region of S-gene (recommended to be tested simultaneously with SARS-CoV-2 diagnostic PCR) should be taken as symptom of presumptive Alpha variant and the sample should be investigated by WGS. By using criteria along these lines, the diversity described may be bypassed to sequences containing the mutation responsible for the S-gene PCR-negative result, ignoring other sequences probably belonging to Alpha variant.

In the present article, we report a fast, simple, and affordable strategy to identify the most relevant SARS-CoV-2 lineages described so far. The method is based on the Sanger-sequencing of a 921 bp S-gene fragment comprising 30 informative variable sites rather than these based on the complete or almost complete sequencing of the S-gene or whole-genome sequencing, as previously reported [21,22,23,24]. The strategy proposed in the present study can be performed with only a single PCR and a simple analysis, constituting a cost-effective methodology for most diagnostic laboratories. The reagents and protocol have been made available world-wide by Genetic PCR Solutions™ (GPS™, Alicante, Spain), also providing a simple graphical user interface (GUI) and a reference database including reference genomes from all variants concerned.

## 2. Materials and Methods

### 2.1. Determination of Informative Nucleotide Positions from Genome Analysis

Ten complete genomes of each variant, namely VOC Omicron and Delta, VBM Alpha, Beta, Gamma, Epsilon, Eta, Iota, Kappa, Lambda, and Mu, and lineages B.1.617.3, A.23.1, C.16, B.1.258, and B.1.1.207 were downloaded from the GISAID database and evaluated to determine the presence of mutations. Different genomes from separate laboratories at diverse world regions, excluding redundant data, not complete data, and low-coverage genomes, were selected to maximize diversity for each variant. Individual genes (ORF1ab, S, 3a, E, M, Orf6, Orf7a, Orf7b, Orf8, N, and Orf10) were identified on the alignment by comparison with the reference genome (NC_045512.2) retrieved from the National Center for Biotechnology information (NCBI) website databases (Bethesda, MD, USA). Multiple sequence alignment was performed with ClustalW algorithm, and the resulting alignments were analysed for identification of variant-specific nucleotides by using the Phylogenetic Analysis option on the Mega 11.0.10 software [25]. Only single-nucleotide mutations involving amino acid changes were considered (Table A1). Furthermore, selected mutations were thoroughly evaluated for each variant using all available sequences at GISAID database deposited until July 2022 (12,349,470 sequences). In this step of the analysis, not only individual mutations were evaluated but also their combinations: the aim was to determinate variant-specific signature sequences. Subsequently, the emerging Omicron sub-lineages were analysed to evaluate the discrimination capacity of the fragment.

### 2.2. Preparative End-Point Reverse Transcriptase-PCR

Specific sets of primers flanking a selected 921 bp fragment of the S-gene and all reagents used to perform the reverse-transcriptase end-point PCR (RT-epPCR) were supplied in the SARS-CoV- 2 seq-RT-epPCR kit (GPS™, Alicante, Spain) [26,27], following manufacturer’s instructions. RT-epPCR reactions were initially carried out using 10^5^ copies/µL of SARS-CoV-2 synthetic RNA Control 2 Wuhan-Hu-1 (GenBank ID MN908947.3) from Twist Bioscience (South San Francisco, USA). All the PCRs were performed with a final volume of 25 µL, containing 6 µL of GPS™-mix-RT (5×), 1 µL of primer-probe mix, 16 µL of H_2_O, and 2 µL of sample. The one-step amplification regime included 10 min of retrotranscription at 48 °C, an activation step of 2 min at 95 °C, and 35 cycles composed by a denaturation step at 95 °C for 5 s, 30 s at 54 °C for annealing, and 90 s at 72 °C for the extension. A final elongation step was added, lasting 5 min at 72 °C. Samples were analysed in an agarose gel (agarose D1 LOW EEO, Pronadisa) using 4 µL of PCR product and 2 µL of EZVision (VWR Life science) running for 30 min with 115 V.

### 2.3. Partial S-Gene Sequencing and Phylogenetic Analysis

All the samples with a band visible to the naked eye on the agarose gel were purified using GPSpin PCR Kit (GPS™, Alicante, Spain) and eluted with 30 µL of DNase/RNase free water. The amount (ng/µL) of DNA was determined studying 2 µL of the sample in the spectrophotometer DS-11 (Denovix, Wilmington, DE, USA). Between 10–30 ng/µL of each purified PCR product and sequencing primers at 5 µM were mixed with a final volume of 10 µL and submitted to GATC (Eurofins Genomics, Coralville, IA, USA) services for Sanger-based sequencing. Resulting sequences were analysed by comparison with a S-gene sequence reference database. Five S-gene 921 bp fragment sequences of each variant were included into this reference database, corresponding with the most diverse genomes for each variant (Table A2) harvested from the GISAID public database. Alignment and phylogenetic analysis were performed by using the Mega 11.0.10 software [25] with the neighbour-joining method [28] and bootstrap values for 1000 replicates. Alongside this, phylogenetic analysis of obtained sequences and sequences retrieved from the reference database were achieved by using a newly developed program, which is available with a simple graphical user interface (GPS™, Alicante, Spain) following the designer’s instructions.

### 2.4. Analytical Validation

Preliminary experimental assays were carried out with 10^5^ copies/µL of SARS-CoV-2 synthetic RNA Control 2 Wuhan-Hu-1 (GenBank ID MN908947.3) from Twist Bioscience (South San Francisco, CA, USA). Furthermore, analytical validation included the testing of RNA extracts from 129 clinical SARS-CoV-2-positive samples provided by two external clinical laboratories. It was confirmed by using the GPS™ COVID-19 dtec-RT-qPCR kit (Alicante, Spain).

## 3. Results

### 3.1. In Silico Specificity of the S-Gene 921 BP-Fragment

The sequence analysis of ten complete genomes of each variant, i.e., Omicron, Delta, Alpha, Beta, Gamma, Epsilon, Eta, Iota, Kappa, Lambda, and Mu, and lineages B.1.617.3, A.23.1, C.16, B.1.258, and B.1.1.207 (GISAID) confirmed that S-gene concentrates a considerable number of mutations compared to other parts of the genome. The fragment of 921 bp, ranging from positions 22,784–23,705 of the reference genome Wuhan-Hu-1 (NCBI ref.: NC_045512.2) comprised 30 variable nucleotide positions whose composition was considerably specific for the variants under analysis. All 30 mutations of each variant were evaluated against 12,250,005 genomes (GISAID, July 2022); afterwards, the frequencies were calculated (shown in Appendix A). Finally, the combination of these mutations yielded a high level of in silico inclusiveness (proportion of genome sequences of a determined variant containing the corresponding specific mutation pattern) and exclusiveness (% of sequences of a variant containing a specific mutation pattern compared to all SARS-CoV-2 genomes containing the specified pattern) for all the variants subjected to this search (Table 1). Therefore, the sequence of this 921-fragment was considered informative to determine SARS-CoV-2 variants and was subjected to phylogenetic analysis. The resulting neighbour-joining phylogenetic tree obtained using a simplified database containing five S-gene 921 bp fragment sequences for each variant (Table A2) showed clearly separated clusters for each SARS-CoV-2 variant (Figure 1). With this methodology, Omicron sub-lineages BA.4 and BA.5 as well as XE and BA.2 could not be differentiated from each other although all of them fell within the cluster corresponding to the Omicron variant.

### 3.2. In Vitro PCR-Amplification, Sequencing, and Phylogenetic Analysis of the Assay

Optimization of the RT-epPCR protocol was performed by selecting the most suitable set of primers, which is supplied in the SARS-CoV- 2 seq-RT-epPCR kit (GPS™, Alicante, Spain) and 10^5^ copies/µL of reference SARS-CoV-2 synthetic RNA Control 2 Wuhan-Hu-1 (GenBank ID MN908947.3). The optimized RT-epPCR protocol was applied to purified RNA from clinical samples previously determined as SARS-CoV-2-positive and confirmed in our laboratory by real-time PCR. Amplified PCR products were analysed by agarose electrophoresis, and the expected bands corresponding to the 921 bp fragment size were visualized in an UV-transilluminator. Figure 2 shows the results obtained from clinical samples (2021/SARS-CoV-2/GPS001, 2021/SARS-CoV-2/GPS002, 2022/SARS-CoV-2/GPS003, and 2022/SARS-CoV-2/GPS004). Samples with low RNA concentration (Ct > 29 in qPCR, below 1000 copies) may not yield enough amplicons for sequencing.

Purified products of the RT-epPCR were subjected to sequencing, and the resulting sequence data were processed as previously described using our reference database to obtain a phylogenetic tree for variant determination (Figure 1). In the resulting tree, the sequences from analysed samples clustered with their corresponding variant as shown in Figure 2: sample 2021/SARS-CoV-2/GPS001 clustered with the B.1.1.7 lineage (variant Alpha); sample 2021/SARS-CoV-2/GPS002 with B.1.617.2 lineage (variant Delta); and samples 2022/SARS-CoV-2/GPS003 and 2022/SARS-CoV-2/GPS004 with cBA.4/BA.5/B.1.529 lineages (variant Omicron). The same result was obtained when using a simple graphical user interface recently developed (GPS™, Alicante, Spain) that automatically assembles, aligns, and generates a phylogenetic tree.

### 3.3. Analytical Validation of the Method

The developed method for SARS-CoV-2 variant identification was applied to RNA extracted from a COVID-19-positive clinical samples received by our laboratory (Table 2) and corroborated in our laboratory by real-time PCR (not shown). Samples collected during early period April–June 2021 showed a clear tendency to be identified as Alpha variant. This trend changed throughout July, with a clear twist towards the Delta variant, which reached its pinnacle during August 2021. A few additional samples collected in 2022 were identified as Omicron variant. Finally, five samples could not be assigned to any variant by the assay, representing 3.9% of samples analysed (5/129), as they may belong to other variants not included in our database.

## 4. Discussion

A considerable number of SARS-CoV-2 variants have emerged during the last two years, but only a few acquired mutations affect the pathogenic characteristics of the virus. Among all of them, the Omicron (lineages B.1.1.529 and BA.x), Delta (lineages B.1.617.2 and AY.x), and Alpha (lineages B.1.1.7 and Q.x) have spread worldwide to such point they represent ca. 90% of all genomes deposited into GISAID. Current lineage identification has been addressed by two main strategies: qPCR-based targeting of single-nucleotide mutations or whole-genome sequencing. In our laboratory, a review of some single-nucleotide mutations of SARS-CoV-2 (∆69–70, N501Y, and ∆144) revealed that they were not always variant-specific as expected. For instance, mutation ∆69–70 has been largely considered exclusive for Alpha variant; however, by February 2021, only 82% of known SARS-CoV-2 genomes containing mutation ∆69–70 were Alpha, while the remaining 18% corresponded to several other variants (data not shown). Afterwards, the number of Alpha genomes increased exponentially, which contributed to artificially increase the degree of in silico specificity of the mutation. Consequently, a proper lineage identification required the simultaneous analysis of multiple single-nucleotide mutations, making the qPCR approach more complex. It is generally accepted that whole-genome sequencing (WGS) is considered as the gold-standard method to determine SARS-CoV-2 variants. Nevertheless, this technology is complex and expensive, not affordable for most laboratories; therefore, an artificial criterion to identify the Alpha variant (B.1.1.7) was stablished to select the samples of interest for WGS in well-equipped reference laboratories. The protocol was named Spike Gene Target Failure (SGTF) and was based in a qPCR targeting ∆69–70 of the S-gene, assayed simultaneously with SARS-CoV-2 diagnostic PCR. According to this protocol, a negative result in the qPCR targeting ∆69–70 should be considered as presumptive Alpha variant, and the sample must be submitted to WGS analysis. Unfortunately, this criterion generated a biased enrichment of ∆69–70 Alpha genome sequences on databases and could have masked other populations of minority lineages.

This new strategy for SARS-CoV-2 variant identification based on partial S-gene sequencing may solve the issues derived from the information provided by single-nucleotide mutation PCR, also avoiding the complexity/expenses of WGS. The simplicity makes the method described here a candidate tool to be implemented in laboratories of control, without a need for outstanding skills and investments. Results revealed that the presence of a specific combination of mutations in the selected short fragment allowed the identification of several relevant variants, including VOC Delta and Omicron and VBM Alpha, Beta, Gamma, Eta, and Lambda, with high levels of inclusivity and exclusivity (Table 1).

Although all sub-lineages of Omicron were grouped in a single cluster including the first lineage B.1.1.529, some of them showed a very close phylogenetic relationship and could not be differentiated form each other, for instance, BA.4 and BA.5 as well as XE and BA.2. The identification results retrieved from the RNA extracts of 129 SARS-CoV-2-positive clinical samples revealed a good correlation with the predominant variant circulating during the time of sampling, as the collection was performed through several COVID-19 waves. The Ministry of Health of Spain has reported predominance of the Alpha variant since the beginning of 2021, an increase in the Delta variant from July 2021 to December 2021, and the emergence of the Omicron variant during January 2022.

Controlling the spread of emergent variants could be crucial in the fight against SARS-CoV-2. The assay developed and described in the present article, available from GPS™ (Alicante, Spain), allows a quick and affordable identification of the main variants catalogued as VOC/VOI by competent entities. This tool could provide resources to laboratories that do not have the technological capacity to undertake WGS, without the need to restrict the number of samples through previous criteria, hence being a massive contribution.

## 5. Conclusions

The appearance of new viral variants, such as those of SARS-CoV-2, is a natural viral process, which can lead to diagnosis failures and treatment modifications. The possibility to easily track lineages would shed light into the spread of the SARS-CoV-2 disease but also in the monitoring of VOC, VOI, and VBM through their expansion, alerting authorities and health care workers of new possible outbreaks. This developed strategy for SARS-CoV-2 variant identification based on partial S-gene sequencing could simplify virus monitoring since it is a fast and reliable strategy to evaluate every SARS-CoV-2-positive sample detected through a manageable and affordable analysis.

## Figures and Tables

**Figure 1 viruses-14-02588-f001:**
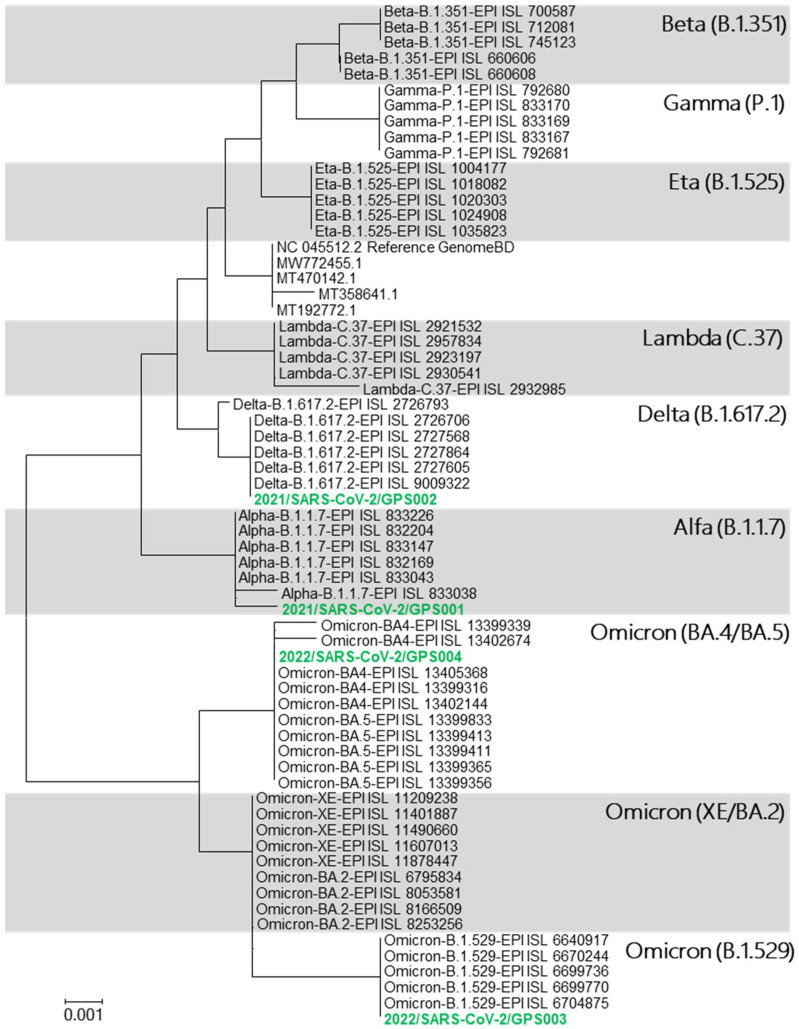
Phylogenetic tree displaying the relationships between the SARS-CoV-2 lineages based on the selected S-gene 921 bp fragment database. It also shows the phylogenetic affiliation of the four clinical samples (green) assayed.

**Figure 2 viruses-14-02588-f002:**
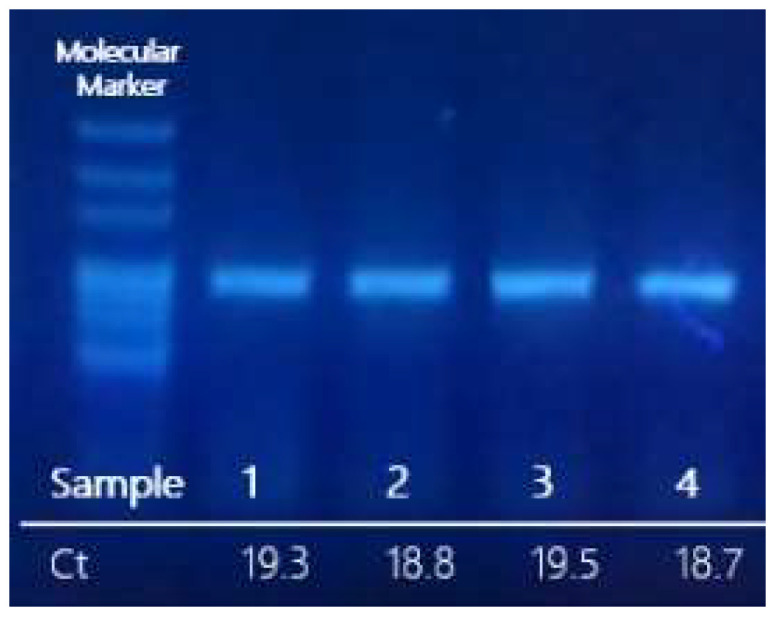
Agarose gel of the preparative RT-epPCR products from SARS-CoV-2-positive RNA clinical samples. The Ct obtained by qPCR (GPS™ SARS-CoV-2 dtec-RT-qPCR) assay has been indicated below each lane.

**Table 1 viruses-14-02588-t001:** Informative amino acid mutations and degree of in silico specificity of the S-gene 921 bp fragment (inclusiveness and exclusiveness) for variant identification.

SARS-CoV-2 Variants	Informative Mutations	Inclusivity	Exclusivity
ALPHA B.1.1.7+Q.x	N501Y+A570D+D614G+P681H	96.9%	97.1%
BETA B.1.351+B.1.351.2+B.1.351.3	K417N+E484K+N501Y+D614G+A701V	79.0%	93.0%
GAMMA P.1+P.1.x+P.1+P.1.x.x	K417T+E484K+N501Y+D614G+H655Y	92.2%	99.8%
ETA B.1.525	E484K+D614G+Q677H	98.0%	85.4%
DELTA B.1.617.2+AY.x+AY.x	L452R+T478K+D614G+P681R	98.0%	99.4%
LAMBDA C.37+C.37.1	L452Q+F490S+D614G	97.3%	90.1%
OMICRON B.1.1.529	S477N+E484A+T478K+Q493R+G496S+Q498R+N501Y+Y505H+T547K+D614G+H655Y+N679K+P681H	86.3%	98.8%
OMICRON BA.4+BA.5	K417N+N440K+L452R+S477N+T478K+E484A+F486V+Q498R+N501Y+Y505H+D614G+H655Y+N679K+P681H	89.9%	89.3%

**Table 2 viruses-14-02588-t002:** SARS-CoV-2 variant identification by phylogenetic analysis of the selected S-gene 921 bp fragment determined from the RNA extracts of 129 clinical samples positive for COVID-19.

Sampling Date	Number of Samples	Alpha (B.1.1.7)	Delta (B.1.617.2)	Gamma (P.1)	Omicron (B.1.529+BA.X)	U/I Variant
23-04-21	1	1	0	0	0	0
27-04-21	2	2	0	0	0	0
28-04-21	1	1	0	0	0	0
18-06-21	8	6	0	0	0	2
21-06-21	3	3	0	0	0	0
22-06-21	4	4	0	0	0	0
02-07-21	3	0	3	0	0	0
03-07-21	2	0	1	0	0	1
05-07-21	7	1	5	1	0	0
06-07-21	9	0	8	0	0	1
07-07-21	2	1	1	0	0	0
08-07-21	7	0	6	0	0	1
10-07-21	6	0	6	0	0	0
12-07-21	3	0	3	0	0	0
13-07-21	2	0	2	0	0	0
17-07-01	10	0	10	0	0	0
19-07-21	21	0	21	0	0	0
20-07-21	33	2	31	0	0	0
13-08-21	1	0	1	0	0	0
04-02-22	2	0	0	0	2	0
17-05-22	2	0	0	0	2	0
TOTAL	129	21	98	1	4	5

## Data Availability

All data have been included in the text.

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
