# Peer review of "SARS-CoV-2 Variants Identification; A Fast and Affordable Strategy Based on Partial S-Gene Targeted PCR Sequencing"

_viruses, 2022, doi:10.3390/v14112588_

Round 1

Reviewer 1 Report

Dear authors ,

in this article, you report a new and fast strategy to identify the most relevant SARS-CoV-2 lineages described so far. The method is based on the Sanger-sequencing of a 921 bp S-gene fragment comprising 30 informative variable sites and following comparison with the already known sequences. I'm not sure that analyzing just a partial-S-gene sequencing can discriminate between the variants. Nonetheless, the manuscript is clear and well written. I have only a few corrections to suggest.

- Line 36-37: Insert a table with a list of VOCs, VOIs and VBMs or insert a reference to access (es.https://www.ecdc.europa.eu/en/covid-19/variants-concern)
- Line 81 and 143: Which of the Omicron variants was used for your study? Indicate it, please

After minor revision, the manuscript can published for me.  Best regards. 

Reviewer 2 Report

Since the emergence of omicron mutation since December 2021, omicron mutation has been reported in a wide variety. Variational identification and analysis through simple S gene analysis is not a new method at this time. In order to check the mutation, it is considered that the usefulness is low at the moment when it is necessary to check the variation of other regions other than the S gene.

The page 2, Line 92, and MEGA5.2.2 programs are very old. You must use the latest version, MEGA-X or later.

Please modify page 6, Table 2, and Alpha.

In Page 7, Line 237, variants analysis using the S gene is not a new strategy. Since many papers have been reported, it should be explained by a mutation search technique with good cost benefits rather than new.

In Page 7, Line 241, for various variants and recombinant variants since the emergence of the omicron variant, it is an error to state that everything is possible in the present where various variants other than the S gene are reported.

Round 2

Reviewer 2 Report

It's been reinforced a lot for the parts that need to be modified. Although the method of analyzing the S gene to identify the mutation is not very new, the progress of the paper and the results of the sample provided sufficient evidence as a paper.